# The Landscape of Accessible Chromatin and Developmental Transcriptome Maps Reveal a Genetic Mechanism of Skeletal Muscle Development in Pigs

**DOI:** 10.3390/ijms24076413

**Published:** 2023-03-29

**Authors:** Lingli Feng, Jinglei Si, Jingwei Yue, Mingwei Zhao, Wenjing Qi, Siran Zhu, Jiayuan Mo, Lixian Wang, Ganqiu Lan, Jing Liang

**Affiliations:** 1Laboratory of Animal Genetics and Breeding, College of Animal Science and Technology, Guangxi University, Nanning 530004, Chinagqlan@gxu.edu.cn (G.L.); 2Institute of Animal Science, Chinese Academy of Agricultural Sciences, Beijing 100097, China

**Keywords:** ATAC-seq, transcriptome sequencing, skeletal muscle, muscle fiber development, pig

## Abstract

The epigenetic regulation mechanism of porcine skeletal muscle development relies on the openness of chromatin and is also precisely regulated by transcriptional machinery. However, fewer studies have exploited the temporal changes in gene expression and the landscape of accessible chromatin to reveal the underlying molecular mechanisms controlling muscle development. To address this, skeletal muscle biopsy samples were taken from Landrace pigs at days 0 (D0), 60 (D60), 120 (D120), and 180 (D180) after birth and were then analyzed using RNA-seq and ATAC-seq. The RNA-seq analysis identified 8554 effective differential genes, among which *ACBD7*, *TMEM220,* and *ATP1A2* were identified as key genes related to the development of porcine skeletal muscle. Some potential cis-regulatory elements identified by ATAC-seq analysis contain binding sites for many transcription factors, including SP1 and EGR1, which are also the predicted transcription factors regulating the expression of *ACBD7* genes. Moreover, the omics analyses revealed regulatory regions that become ectopically active after birth during porcine skeletal muscle development after birth and identified 151,245, 53,435, 30,494, and 40,911 peaks. The enriched functional elements are related to the cell cycle, muscle development, and lipid metabolism. In summary, comprehensive high-resolution gene expression maps were developed for the transcriptome and accessible chromatin during postnatal skeletal muscle development in pigs.

## 1. Introduction

The domestic pig (*Sus scrofa*) is one of the main protein sources utilized by humans and an excellent research model for human muscle-related diseases [1,2]. The association between muscle insulin resistance and mitochondrial dysfunction is a common feature in the elderly and people who are susceptible to or have dominant type 2 diabetes mellitus [3,4]. The skeletal muscle is mainly composed of connective tissue and muscle fibers, which have the ability to regenerate and self-repair after injury [5,6]. Muscle fiber traits mainly include muscle fiber density, size, and type [7,8]. Skeletal muscle development during embryonic and postnatal development is a paradigm of stem and progenitor cell maintenance, lineage differentiation, and terminal differentiation [9,10]. Specifically, these include myogenesis in the embryonic stage and the determination of the number of muscle fibers [11,12], as well as the hypertrophy of skeletal muscle cells and the transformation of muscle fiber types after birth [13,14,15]. Embryonic skeletal muscle in the early stage of development is most widely studied, and this has resulted in genome-wide transcriptional profiles for porcine embryonic skeletal muscle (33, 65, and 90 days of gestation) for both Chinese and foreign breeds, and these were analyzed using LongSAGE technology [16]. However, there were relatively few studies on pig postnatal skeletal muscle. In addition, a fine map of the open regions of the chromatin in the skeletal muscle genome throughout the different developmental stages of the pig embryo has also been mapped [17]. However, there are even fewer studies on the landscape of accessible chromatin in pig postnatal skeletal muscle development. Many aspects of postnatal muscle development are similar to the embryonic developmental process [18,19], and studies of all aspects of myogenesis are developed to gain a comprehensive understanding of the required muscle formation. Therefore, it is of great significance to explore the regular mechanisms of postnatal skeletal muscle development in pigs.

Myogenesis is a complex biological process associated with many changes in mammalian gene expression [20]. Myogenic regulators such as *MyoD*, *MyoG*, *Myf5*, and *Myf6* co-regulate the expression of muscle-specific genes and control skeletal muscle development [21]. Paired box transcription factors (PAX) play key roles in the multiple stages of muscle development, such as Pax7 and Pax3, the most reported members of the Pax family, which are very important transcription factors and are expressed in satellite cells [22,23]. They also play a regulatory role in self-renewal and cell proliferation. Moreover, the *MYH3* gene can affect muscle fiber type composition and intramuscular fat content in pigs [24,25], and myostatin (*MSTN*) can negatively regulate skeletal muscle growth and development through autocrine or paracrine signaling [13,26]. In addition, skeletal muscle development is associated with epigenetic regulatory mechanisms [27]. For example, DNA methylation dynamically regulates skeletal muscle development by affecting the expression of muscle-related genes by regulating the binding of upstream myogenic transcription factors [28], and histone methylation modification can participate in skeletal muscle growth by changing the structure of the nucleosomes and interacting with other epigenetic factors [29]. However, relatively few reports have exploited genome-wide chromatin accessibility to regulate porcine skeletal muscle growth and development through the binding of different transcription factors [17,18]. Furthermore, the dynamic changes in chromatin accessibility and gene expression during postnatal muscle growth and development in pigs at the epigenetic and transcriptional levels have not yet been systematically described.

The genetic information of eukaryotes is stored in the form of chromatin [30], and the basic unit of chromatin is the nucleosome [31]. A nucleosome consists of two histone H2A, two histone H2B, two histone H3, two histone H4 octamers, and 146 bp of DNA wrapped around the outside [32,33]. Accessible chromatin is known to mark regulatory sequences, including promoters, enhancers, and silencers, and it can dynamically control gene expression through interactions with transcription factors (TFs) [34]. An assay for transposase accessible chromatin with high-throughput sequencing (ATAC-seq) is currently the preferred technical method of choice for studying epigenomics [35,36]. At present, the differences in molecular mechanisms during the postnatal period of porcine skeletal muscle development have not been widely analyzed in terms of chromatin openness characteristics. Through the combination of ATAC-seq and RNA-seq data, we can identify some of the underlying core genes that regulate muscle development and the pathways that regulate these processes [37]. Therefore, our study aimed at identifying key genes and regulatory pathways involved in *Longissimus dorsi* muscle development.

Here, we have performed ATAC-seq and RNA-seq in the live tissues of porcine *Longissimus dorsi* muscle at different postnatal stages (0, 60, 120, and 180 days) for the same Landrace pigs. Through bioinformatics analysis and experimental verification, we found that chromatin accessibility was related to the expression of the target gene *ACBD7* in the TGF-beta pathway during pig skeletal muscle development and identified other key pathways and genes involved in pig skeletal muscle development after birth. In summary, we have partially identified important regulatory factors and epigenetic regulatory mechanisms for postnatal skeletal muscle development in pigs, which will help to further analyze the molecular regulatory network of muscle growth and development in the future.

## 2. Results

### 2.1. Data Filtering, Alignment Analysis, and Expression Statistics for RNA-Seq 

The total number of bases obtained by data filtering statistics was 6,601,591,800 (Figure 1A). The raw data were quality controlled (N ratio > 5%) (Figure 1B) and filtered, resulting in an average of 44 million reads for each library (Appendix A). Hisat2-2.2.1 software was used to align clean reads with a pig reference genome (*Sus scrofa* 11.1 genome), and the results showed that 96% of the overall mapping was observed in samples from all age groups, while 85–90% of the reads were uniquely mapped. The number of sequences aligned to exons, introns, and intergenic regions was counted, and a bar plot was made according to the ratio (Figure 1C). Most of the sequences were mapped to the exon region, indicating that the annotation information for this species was relatively comprehensive.

ASprofile (version 1.0.4) software was used to analyze and count the alternative splicing events of each sample based on the known gene model (Figure 1D). Compared with the known genome, the percentage of each event, such as cassette exons (MSKIP), retention of multiple introns (MIR), transcription start site (TSS), alternative exon ends (AE), and skipped exons (SKIP), was determined according to the gene expression levels for all samples (Figure 1E and Appendix A) to identify the overall trends in the sample expression levels. The results of the dendrogram show that the biological replication and classification of each tissue cluster were scientific (Appendix A). The correlation between samples was analyzed using the principal component analysis (PCA) (Figure 1F). The heatmap showed the distribution of gene expression levels in each sample (Figure 1G). The Pearson correlation coefficient between samples with similar genetic backgrounds was >0.87, which further demonstrated that the classification of the samples submitted for inspection was reasonable and the influence of batch effects on data analysis was removed (Figure 1H). Taken together, the results show that the generated RNA-seq sequencing data were reliable.

### 2.2. Enrichment Analysis of Differentially Expressed Genes during Postnatal Skeletal Muscle Growth 

Based on the criteria for screening differentially expressed genes|log2 (Fold Change)| ≥ 1 and FDR < 0.05, the 8554 effective differential genes were obtained using the DESeq2 R package (1.30.1). After pairwise comparison of the four periods, the differential gene statistics were performed on the groups that were up- and down-regulated (Figure 2A,B). Among them, the expression of the *ACBD7* and *TMEM220* genes screened in the up-regulated group increased continuously with the growth of porcine skeletal muscle. 

The differential genes between the four stages of skeletal muscle development were divided into three types for analysis. For the D60 vs. D0, D120 vs. D0, and D180 vs. D0 groups, 1558 DEGs were subjected to functional enrichment analysis (Appendix A). The top 10 most enriched GO entries based on up-regulated genes and down-regulated genes (Figure 2D,G). Additionally, in the D180 vs. D60 and D120 vs. D60 groups, 153 genes were subjected to functional enrichment analysis (Appendix A), and the six most enriched GO entries in relation to up-regulated genes and the nine most enriched GO terms associated with down-regulated genes were displayed (Figure 2E,H). Moreover, for the three groups D60 vs. D0, D120 vs. D60, and D180 vs. D120, there were 12 differentially co-expressed genes, and they were mainly associated with the TGF-beta signaling pathway (*p* < 0.05) (Appendix A). The DEGs-enriched KEGG pathways containing up- and down-regulated pathways between different developmental stages are shown in Figure 2F,I.

### 2.3. Trend Analysis for All RNA-Seq Genes, Cluster Analysis, and Gene Ontology 

To determine the trend in the changes in the different stages of skeletal muscle development, the median values from the three biological replicates of the skeletal muscles were imported into STEM tools to conduct sequential analysis of the RNA-seq at different developmental stages in the Landrace pigs. As a result, a total of 50 profiles were obtained (Figure 3A). At the same time, the expression level data obtained from the RNA-seq in the previous step was organized into a table and then trended using the Mffuz package to obtain the expression trends of six different genes during the four different stages of skeletal muscle development in Landrace pigs (Figure 3B). 

The modules with similar trends were then subjected to functional enrichment analysis. The gene sets in profile 12, 1, 9, and 0 were mainly enriched in angiogenesis, skeletal development, cardiac development, and positive regulation of smooth muscle (Appendix A). The gene sets in profiles 42, 48, and 49 were mainly enriched in cell-cell matrix adhesion, G2/M mitotic cell cycle, and negative regulation of protein ubiquitin (Appendix A). The gene sets in profiles 3 and 4 were mainly enriched in smoothened signaling pathways, positive regulation of synapse assembly, and negative regulation of transforming growth factor beta receptors (Appendix A). Genes in profile 11 are mainly enriched in receptor-mediated endocytosis, the negative regulation of protein kinase activity, and neuronal projection morphogenesis, whereas the genes in profile 2 are mainly enriched in the positive regulation of epithelial cell proliferation and lung regulation of developmental apoptosis. The genes in profile 47 are mainly enriched in antigen processing and presentation of peptide MHC Class Ⅱ, chemokine-mediated signaling, and the inflammatory response (Appendix A).

Among the results analyzed using the Mffuz package, GO enrichment analysis was performed on the six clusters obtained. Cluster 1 and Cluster 2 showed a trend of rising and then falling, and the functions were mainly enriched in muscle cell migration, chromosome segregation, and negative regulation of muscle cell differentiation (Appendix A). The gene set of Cluster 3 was first decreased and then increased, and the function was mainly enriched in regulation of osteoblast differentiation, negative regulation of the Wnt signaling pathway, and skeletal muscle cell differentiation (Appendix A). The genes in Cluster 4 were mainly enriched in intracellular transport, organic nitrile complex catabolism, protein catabolism, and the glycerophospholipid biosynthesis signaling pathway (Figure 3C). Cluster 5 and Cluster 6 showed a downward trend, and the genes were mainly enriched in fibroblast growth factor-stimulated cell response, Notch signaling pathway, striated muscle tissue development, muscle structure tissue development, and regulation of cellular response to growth factor stimulus (Figure 3D, Appendix A). The significant pathway maps were then screened using the KEGG enrichment in the six trend clusters (Figure 3E). The main enriched pathways were the chemokine, mTOR, MAPK, cGMP-PKG, PI3K/AKT, TGF-beta, PPAR, and VEGF signaling pathways. Overall, trend analysis was performed, and it was found that functions were mainly enriched in angiogenesis, chemokine-mediated signaling, mast cell activation, smooth muscle cell differentiation, growth factor response, the Notch signaling pathway, the development of striated muscle tissue, and the regulation of cardiomyocyte proliferation.

### 2.4. ATAC-Seq Read Mapping, Processing, and Visualization

The genomic distribution of the open chromatin peaks across all muscle tissue samples was assessed using the assay for transposase-accessible chromatin through sequencing (ATAC-seq). We performed ATAC-seq on skeletal muscle tissue. First, Trimmomatic (version 0.39) software was used to remove the joints, and the low-quality data was filtered to obtain the original data. ATAC-Seq clean reads were aligned to the *Sus scrofa* 11.1 genome using Bowtie2 (version 2.4.2) default parameters. More than 165 million reads were obtained for each biological replicate through paired-end sequencing, and an average of 94.69% of all reads from skeletal muscle tissue were successfully mapped. All ATAC-seq samples had over 111 million effective reads, which surpassed the ENCODE project standard of 50 million effective reads for ATAC-seq data (https://www.encodeproject.org/atac-seq/ (accessed on 2 February 2022)). Acceptable percentage of the mapped reads according to the ENCODE ATAC-seq standard (alignment rate > 80%) (Appendix A). 

In addition to the length distribution of the transposase-cut fragments (Figure 4A), the transcription start site (TSS) enrichment analyses (Figure 4B) and the consistency of peaks between replicates were evaluated using IDR, which showed that our ATAC-seq data were qualified for further analyses (Figure 4C). The distribution of these peaks for each chromosome can be viewed globally using the function “covplot” (peak, weightCol = “V5”) (Figure 4D). The coverage area of the peak on the chromosome was displayed and visualized. The “computeMatrix”, “plotHeatmap”, and “plotProfile” functions in the deepTools package were used to generate a heat map displaying the ATAC-seq data (Figure 4E). Each row in the heatmap is a peak, and different colors represent different amounts of enriched reads. It can be seen that the red color at 0 days of age is the deepest, which verifies the speculation that the peaks of 0-day-old pig skeletal muscle are the most and the chromatin accessibility is more active than the other three periods.

### 2.5. Enrichment Analysis of nearby Genes of Differential Open Chromatin Regions

Many different chromatin states were defined based on epigenetic markers of skeletal muscle tissue in Landrace pigs. Most peaks were annotated to intergenic regions, followed by intronic and promoter regions (Figure 5A). To further elucidate the regulatory roles of stage-specific ACR-annotated genes during embryonic skeletal muscle development, the annotated genes were analyzed through functional enrichment. At the D0, D60, D120, and D180 development stages, there were 553 stage-shared ACRs, and their annotated genes were subjected to functional enrichment analysis (Figure 5B). The results of the GO enrichment analysis showed that the 553 ACRs were significantly enriched in 93 GO entries, including 43 biological processes (BPs), 32 molecular functions (MFs), and 18 cellular components (cellular component, CC); the KEGG enrichment analysis showed that there were 38 entries, and the top five were screened as shown in Figure 5D. They were mainly related to ubiquitin protein ligase activity, Wnt, FoxO, and mTOR signaling pathways (Figure 5D). The different ACRs detected in each stage were annotated into the genome; the difference ACRs during muscle development at the D60 vs. D0, D120 vs. D0, D180 vs. D0, D120 vs. D60, D180 vs. D60, and D180 vs. D120 stages were annotated as having 756, 1492, 665, 569, 15, and 46 genes, respectively (Figure 5C). The enriched KEGG entries in each group are 19, 18, 21, 10, 6, and 18, respectively. The top six pathways in muscle development mainly include the TGF-beta, PI3K-Akt, cGMP-PKG, FoxO, and AMPK signaling pathway, insulin resistance, adrenergic signaling in cardiomyocytes, and the Wnt signaling pathway (Figure 5E).

### 2.6. Motif Analysis of Differential Peaks during Postnatal Porcine Skeletal Muscle Development

The access of transcription factors (TFs) to cis-regulatory elements, such as promoters and enhancers, was limited by the unopened chromatin in the nucleus. The peak-calling function of the HOMER package was used to identify open chromatin regions, which likely contain TF binding sites that can recruit TFs to regulate the expression of nearby genes. The motifs at D0, D60, D120, and D180 during postnatal skeletal muscle development in pigs were differentially analyzed, and 59 co-differential motifs were screened (*p* < 0.05) (Figure 6A). Based on *p*-values between D0 vs. D60, D0 vs. D120, D0 vs. D120, D60 vs. D120, D60 vs. D180, and D120 vs. D180 groups, the top five transcription factor binding motifs were displayed and shown to be enriched in significant peak regions (Figure 6B–D). Among the 59 co-differentiated motifs, *SP2* and *EGR1* were significantly enriched (FDR < 0.05) and thus regarded as key regulators.

### 2.7. Relationship between Chromatin Accessibility and Gene Expression in Muscle Tissues 

To explore whether changes in open chromatin regions in ATAC-seq correlate with changes in gene expression, we integrated ATAC-seq data with RNA-seq data. First, in the D60 vs. D0_up, D120 vs. D0_up, D180 vs. D0_up, D120 vs. D60_up, D180 vs. D60_up, and D180 vs. D120_up groups, the significantly up-regulated genes screened through RNA-seq and the significantly open ATAC-seq peaks corresponded to 32, 106, 75, 13, 3, and 1 overlapping genes, respectively. Furthermore, significantly down-regulated genes identified by RNA-seq overlapped with nearby genes with significantly reduced ATAC-seq peaks in the D60 vs. D0_down, D120 vs. D0_down, D180 vs. D0_down, D120 vs. D60_down, D180 vs. D60_down, and D180 vs. D120_down groups, which corresponded to 29, 193, 153, 6, 4, and 5 overlapping genes (Figure 7A). Afterwards, GO and KEGG enrichment analyses were performed on the obtained overlapping genes, respectively (Appendix A). According to GO annotation and KEGG annotation of the DEGs, these up-regulated genes were significantly enriched in pathways related to skeletal muscle cell differentiation, insulin response, positive regulation of vascular endothelial cell migration, and regulation of protein stability. The genes involved included *DAPK1*, *PRKD1*, *TRIM24*, *NEDD4L*, *EGR1*, and *EGR2*. At the same time, these down-regulated genes were significantly enriched in the pathways for cell-cell adhesion and apoptosis, and the involved genes included *CYFIP2*, *DLG4*, *NTN1*, *CCN1*, *EGR1*, *FOS*, *HES1,* and *RHOB*. 

To further determine the relationship between chromatin openness and gene expression and to investigate how transcription factors regulate downstream genes, we used IGV (version 2.11.9) software to analyze the relationship between gene openness and gene expression. The transcription level of the candidate gene *ACBD7* gradually increased from 0 to 180 days, while the chromatin accessibility increased, decreased, and then increased again. It is speculated that this may be due to the involvement of transcription factors or transcriptional repressors in the regulation of gene expression (Figure 7B). Similarly, the candidate gene *TMEM220* showed the same trend (Figure 7C). Different transcriptional programs and biological processes are activated or repressed at specific stages. In addition, in the important regulator region of the gene *MYOD1* that affects muscle development, from D0 to D180, chromatin is less accessible, and the peak intensity at the D0 stage is significantly higher than in other stages (*p* < 0.05) (Figure 7D). The candidate gene *ATP1A2* is associated with the cGMP-PKG signaling pathway, and its transcription level and chromatin accessibility gradually decreased from 0 to 180 days (Figure 7E).

### 2.8. Validation of Screened Candidate Genes

The *ACBD7*, *TMEM220*, *THBS1*, *RHOB*, *PLK3*, *MYL9*, *LRTM1*, *IL18*, *BMP2*, *HES1*, *FOS*, and *CCN1* candidate genes were screened and analyzed using qRT-PCR. It was found that the results of qRT-PCR and RNA-seq data showed similar gene expression trends, which confirmed the accuracy of RNA-seq data. (Figure 8A). Subsequently, the regulatory effect of the *ACBD7* gene on skeletal muscle satellite cells was further analyzed. The tissue expression profiles were analyzed using the heart, liver, spleen, lung, kidney, brain, back muscle, leg muscle, and fat of the Landrace pigs at 0 and 180 days, and it was found that the expression of the *ACBD7* gene in the back muscle changed with time, as it gradually increased, and the difference was significant (*p* < 0.05) (Appendix A). 

The interference of the *ACBD7* gene was investigated in PK15 cells. Compared with the NC-siRNA group, the expression levels of *ACBD7*-siRNA#1 and *ACBD7*-siRNA#2 were decreased, and the difference was extremely significant (*p* < 0.01). The fluorescence intensity of PK15 cells transfected for 24 h, 48 h, and 72 h was analyzed using Image J (verison 1.8.0) software. It was found that the fluorescence intensity at 72 h was the strongest, but the fluorescence intensity at 48 h was not significantly different from that at 72 h. Combined with the results of the qRT-PCR reaction, 48 h after the cells were transfected was the optimal transfection time. (Figure 8B,C, Appendix A). The *ACBD7* gene was successfully transfected in muscle cells, and the green fluorescence map showed that *ACBD7*-siRNA was effectively transfected. The mRNA levels of *MyOD1*, *PAX7*, and *MYOG*, which are markers for skeletal muscle development, decreased. In addition, the *MyHC1* mRNA decreased, while *MyHC2a*, *MyHC2b*, and *MyHC2X* increased (Figure 8D,E). To validate the role of *ACBD7* in cell proliferation, we overexpressed *ACBD7* in muscle cells. Transwell assays revealed that the *ACBD7* overexpression remarkably increased the cell proliferation abilities of pig SkMCs (Figure 8F,G). The experimental results showed that *ACBD7* may be a key gene in skeletal muscle development regulation. The dynamics of skeletal muscle fiber types at 0, 60, 120, and 180 days were described (Figure 8H). 

## 3. Discussion

The development of skeletal muscle is a complex, dynamic regulatory process that involves a strict timeline and a complex spatiotemporally expressed gene network [38,39,40]. Based on existing epigenetic studies of human and mouse genomes, efforts have been made to generate accurate large-scale predictions for the processes related to DNA methylation, histone modification, RNA methylation, and chromatin accessibility, with a general aim to explore the regulation of genes by TFs in animals [41,42,43]. Studies have suggested that environmental factors may affect offspring through the opening and closing of chromatin [44,45]. The detailed molecular mechanisms of epigenetic inheritance in the pig genome, however, have not yet been fully annotated, especially the regulation of chromatin regions under specific spatial and temporal sequences and active transcription in the genome, which are still being explored [46,47,48]. To fill these knowledge gaps, this study is the first to systematically investigate the chromatin accessibility of *Longissimus dorsi* muscle biopsies taken postnatally from Landrace pigs (after 0, 60, 120, and 180 days) by combining ATAC-seq and RNA-seq analysis. Studying the developmental mechanism of postnatal muscle and the transformation mechanism of muscle fiber types helps to elucidate the molecular mechanisms regulating pig meat production traits. In order to better study the epigenetic modification during the development of postnatal skeletal muscle, it is necessary to reduce the influence of animal genetics, nutrition, temperature, environment, and other factors on skeletal muscle development in pigs. In the study, all Landrace pigs were purebred with the same parents, and 12 muscle samples were collected for biopsies at four time points. The PCA data was analyzed using omics, and we found that the correlation between samples was consistent with the requirements of the ENCODE database, which indicates that the repeatability within the group is better and that there is a good degree of discrimination between the groups. 

In addition, the number of differential genes between D0, D60, D120, and D180 was compared, and day 0 had the highest amount from the volcano plot (Figure 2C). In a previous study, transcriptome analysis and whole-genome resequencing of skeletal muscles from Landrace (lean) and Tongcheng (obese) pigs at 27 time points yielded similar results [49]. Time-series expression miner (STEM) analysis revealed differential gene expression profiles in the Landrace pigs. The TGF-beta, cGMP-PKG, cAMP, and VEGF signaling pathways [50] were identified as important for skeletal muscle development in pigs after birth. The open peaks of the skeletal muscle of the Landrace pigs mostly appeared in the intron and intergenic regions, followed by those around the promoter, which were analyzed using ATAC-seq. The skeletal muscle of the Landrace pigs had the most peaks at 0 days, which indicated that the regulatory elements of the skeletal muscle at 0 days were most active. These data are in line with previous findings in non-model organisms, including pigs, cattle, and sheep, which showed similar patterns for their open chromatin regions [51,52]. ATAC-seq peaks were highly enriched in the promoter area. Our ATAC-seq analysis yielded similar results, and this has improved our understanding of dynamic transcription and chromatin opening in skeletal muscle development.

We further identified the key cis-regulatory elements responsible for modulating the expression of signature genes by overlaying the ATAC-seq results with the RNA-seq data. The largest change in the number of unique peaks during the developmental stage of the Landrace pig’s skeletal muscle occurred between D0 and D60. In addition, the number of emerging and disappearing peaks around candidate key genes varied significantly in the two omics comparisons, and the genomic distribution ratio of these peaks also varied with skeletal muscle developmental stages and gene regulation patterns. We speculated that either the expression of genes is altered following the binding of different transcription factors to their regulatory sites or that differentially accessible sites may not induce any changes in overall RNA expression. The resulting DEGs and qRT-PCR validation results indicated that the key pathway involved the MAPK, P13K-AKt, mTOR, and FoxO signaling pathways [53], lipid metabolism, and Wnt signaling during skeletal muscle development. Among these, the Wnt signaling [54,55] and the mTOR signaling pathways are classic pathways involved in muscle development. These results are similar to those obtained in previous studies [56].

In pigs, the histological profile of the muscle tissue, especially the proportion of individual fiber types, is one of the main factors affecting the meat quality properties [57]. The proportions of MyHC1-type and MyHC2a-type muscle fibers decreased at 60 days, and the proportion of the MyHC2b-type muscle fibers increased sharply. At 60 days postnatal, the proportion of various types of muscle fibers during porcine skeletal muscle development was found to be relatively stable. This was consistent with previous studies, as the transformation process of muscle fiber types is in a dynamic balance as follows: MyHC1 ↔ MyHC2a ↔ MyHC2x ↔ MyHC2b. AMPK/silent information regulator1, and SIRT1/peroxisome proliferator-activated receptor gamma coactivator-1 (peroxisome proliferator-activated receptor γ co-activator 1α, PGC-1α) are important information pathways regulating the transformation of muscle fiber types [57,58]. Various external factors can increase the phosphorylation of *AMPK* and activate its downstream pathways, thereby affecting the expression of the *MyHC* gene and promoting the transformation of muscle fiber types [59]. *IGF1* promotes muscle hypertrophy mainly by enhancing translational initiation and elongation to accelerate protein synthesis in the muscle [60]. Some potential cis-regulatory elements identified by ATAC-seq analysis contain binding sites for many transcription factors, including *SP1* and *EGR1*, which are also the predicted transcription factors regulating the expression of *ACBD7* and *TMEN220* genes. In addition, *EGR1* can promote *MDSC* differentiation through the positive regulation of *MyoG* gene expression [61]. Studies have reported that *ATP1A2* is highly correlated with the electrical conductivity of muscle and is specifically expressed in muscle tissue [62].

To date, most genome-wide chromatin accessibility studies have used cultured or purified cells and frozen tissues [63]. However, in 2015, Greenleaf and Chang et al. combined ATAC-seq with microfluidic technology, and an upgraded version of ATAC-seq, single-cell ATAC-seq (scATAC-seq) [64,65], was gradually applied. Subsequent research in this area may be considered. Here, we generated a new resource consisting of transcriptomics and epigenomics data for porcine skeletal muscle developmental stages and identified key genes and regulatory elements. In the future, we should further explore the network of key genes and regulatory elements regulating skeletal muscle development and analyze its regulatory mechanisms. In conclusion, our study provides a new avenue for understanding the transcriptome and accessible chromatin dynamics during Landrace pigs skeletal muscle development after birth. 

## 4. Materials and Methods

### 4.1. Ethics Statement 

All animal experiments were approved by the Institutional Animal Care and Use Committee of Guangxi University (GXU2018-061).

### 4.2. Animal Samples

ATAC-seq and RNA-seq were performed on the *Longissimus dorsi* muscle tissue of postnatal Landrace pigs from 0, 60, 120, and 180 days. Three full-sib barrows were at 0, 60, 120, and 180 days. Skeletal muscle tissue was live sampled from the same pig at each of the four time points, and three biological replicates were collected at each stage. All Landrace pigs were housed in the same environment and allowed to eat ad libitum during the nursery and grow-finish phases. After sampling, continuous monitoring of the pig’s behavior and the recovery of the pig’s back wound after in vivo sampling showed that the pig’s behavior was normal, and the wound healed completely in approximately two weeks.

### 4.3. RNA Extraction, Library Construction, and RNA Sequencing

The traditional Trizol method [66] was used to extract total RNA from all skeletal muscle tissues, and the integrity of the total RNA was evaluated in 1% agarose. A NanoDrop 2000 was used to assess the concentration and purity of the total RNA. High-quality RNA samples were then used to construct un-stranded, specific libraries. The library was finally sequenced on an Illumina HiSeqXTen (Annoroad Gene Technology Co., Ltd., Beijing, China) as 150-bp paired-end reads.

### 4.4. Alignment of Reads to the Reference Genome for Transcript Assembly and Gene Expression Analysis

A total of 12 samples were used to construct libraries for RNA-seq. The resulting data from library construction and sequencing were converted into raw sequencing reads (sequenced reads) using bcl2fastq2 after the machine was turned off. The raw reads were obtained. The raw data were evaluated for quality using FastQC. Trimmomatic (version 0.39) was used to filter Illumina (FASTQ) sequencing data and remove adapters to reduce the impact on subsequent analyses. The raw data quality control standards were as follows: (1) the sequencing adapters were removed from the reads; (2) reads with an N content of >10% were removed; and (3) single-end sequencing reads that had low-quality (<5) bases in >50% of the read length were removed. Hisat 2 (version 2.2.1) software was used to compare the clean data obtained after quality control with the *Sus scrofa* 11.1 reference genome; read lengths that were not successfully mapped to the reference genome were filtered using the SAMtools (version 1.9.0) software. Stringtie (version 2.2.1) software was used to compare gene assembly transcripts and estimate their expression levels. 

ASprofile software (https://ccb.jhu.edu/software/ASprofile/ (accessed on 10 October 2021)) was used to obtain the alternative splicing type and corresponding expression amount of each sample. The gtf file corresponding to the transcript processed using Stringtie (version 2.1.4) software was analyzed using ASprofile. The DEseq2 package in R (version 3.6.0) software was used for gene difference analysis, and the differences with the gene standards were as follows: FDR < 0.05 and |log2 fold change (FC)| ≥ 1. To avoid false-positive gene expression caused by sequencing bias, genes with low expression levels were removed; all downstream analyses are based on the FPKM ≥ 1 of at least three replicate samples at a certain stage. Time series expression miner (STEM; v1.3.13) analysis revealed the gene expression profiles for the Landrace pigs at 0, 60, 120, and 180 days post-natal. The expression patterns of the genes were analyzed using the Mfuzz R package.

### 4.5. Assay and Library Preparation of Transposase Accessible Chromatin (ATAC)

The ATAC-seq library was prepared according to a frozen tissue improvement protocol previously reported [36]. The ATAC-seq sequencing library can be divided into three steps: nuclear extraction, transposition reaction and purification, and PCR amplification and fragment purification. (1) Preparation of cell samples: the frozen tissue was crushed, approximately 50,000 cells were isolated, and then centrifuged at 4 °C, 500× *g* for 5 min. After this, 50 µL of ice-cold PBS was added to the centrifuge tube from which the supernatant was discarded; it was then washed once and centrifuged at 500× *g* for 5 min at 4 °C. The supernatant was then discarded, and 50 µL of ice-cold lysis buffer suspension was added to the cells in the centrifuge tube, which was then centrifuged at 500× *g* for 5 min at 4 °C. After discarding the supernatant, the transposition was quickly started. (2) Transposition reaction and purification: The treated cells from step (1) were gently placed in a 50 µL transposition reaction system to ensure that the cells were in a floating state. The transposition reaction system used was as follows: TD (2× reaction buffer), 25 µL; TDE1 (Nextera Tn5 Transposase), 2.5 µL; nuclease-free H_2_O, 22.5 µL. Incubated at 37 °C for 30 min, followed by immediate purification of the transposable DNA using the MinElutePCR Purification Kit, and eluted with 10 µL EB. It should be noted that when performing the above operations, the cells were always kept on ice. (3) PCR amplification and fragment purification: the transposed DNA processed in step (2) was subjected to PCR amplification. The PCR reaction system was as follows: transposed DNA, 10 µL; nuclease-free H_2_O, 10 µL; 25 µM Nextera PCR primer 1, 2.5 µL; 25 µM PCR barcoded primer 2, 2.5 µL; NEBNext High-Fidelity 2× PCR Master Mix, 25 µL. The reaction conditions are as follows: 72 °C 5 min; 97 °C 30 s, 97 °C 10 s, 63 °C 30 s, for eight cycles; followed by 72 °C 2 min. PCR amplification was complete, and the MinElute PCR Purification Kit was then used to purify the PCR product, following the manufacturer’s instructions.

### 4.6. ATAC-Seq Library Quality Inspection and Sequencing Data Quality Control and Peak Calling

Before carrying out the transposition reaction, a preliminary quality inspection of the cell sample obtained after the fragmentation was carried out, which involved checking the cell status and the integrity of the nucleus. Only samples with good cell conditions, a clean background, and a relatively complete nucleus were used for downstream library construction. Before sequencing, it is necessary to perform a quality inspection on the constructed library. Usually, the library is quantitatively diluted to 1 ng/µL with Qubit 2.0, and the size of the inserted fragment in the library is detected with an Agilent 2100. Finally, the qualified library was amplified into sequencing clusters using bridge PCR and paired-end sequencing with sequencing read lengths of 150 bp was performed using the Illumina HiSeq × Ten platform.

### 4.7. Mapping and Normalization of ATAC-Seq

Trimmomatic (version 0.39) software was used to remove adaptors, and ATAC-Seq clean reads were aligned to the *Sus scrofa* 11.1 reference genome using Bowtie2 (version 2.4.2) with the default parameters. Mapped reads in .sam format were converted to .bam format and sorted using Samtools 1.9. Mapping reads were filtered using Samtools to keep only those reads with a mapping quality score > 2 (the Samtools “view” command with option “−q 2” to set the mapping quality cutoff). The reads mapped to the mitochondrial genomes were further removed using Samtools (version 1.9) software. After mitochondrial DNA reads were filtered, we obtained properly paired reads using SAMTools. After that, the high mapping quality and qualified reads were further analyzed. Duplicate reads were removed using the Picard Tools MarkDuplicates program (http://broadinstitute.github.io/picard/ (accessed on 14 January 2022)). The R program (version 3.5.2) was used to plot the distribution of the insertion size to evaluate nucleosome and open chromatin integrity. ATAC-Seq peak regions for each sample were called using MACS2 (version 2.2.7.1) with parameters --nomodel --shift −100 --extsize 200. The irreproducible discovery rate (IDR) (https://www.encodeproject.org/software/idr/ (accessed on 8 February 2022)) (version 2.0.0) method was used to evaluate the consistency of the sorted regions/peaks between duplicate samples. In the visualization part, the filtered, sorted, and scaled .bam files were converted to the bigwig format using the “bamcoverage” script in deepTools 2.0 with a bin size of 1 bp, and Reads Per Kilobase Million (RPKM) normalization was performed. Heatmaps and average plots displaying ATAC-seq data were also generated using the “computeMatrix”, “plotHeatmap”, and “plotProfile” functions in the deepTools package are used to analyze heatmaps and the average map of ATAC-seq data. Integrated Genomics Viewer (IGV) 2.11.9 was used to visualize the genome and input it into the bigwig file for analysis.

### 4.8. Analysis of Differential Chromatin Accessibility

We selected the peaks with the strongest correlation with the developmental stage of porcine skeletal muscle as the differentially accessible regions (DARs). The screening threshold for a significant correlation between peak and accessibility was FDR < 0.05. The TF motifs were searched in skeletal muscle tissue-specific peaks using the findMotifsGenome.pl script of the HOMER (version 3.14.0) software (http://homer.ucsd.edu/homer/introduction/instal−l.html (accessed on 16 March 2022)) with the default settings.

### 4.9. Real-Time Fluorescent Quantitative Polymerase Chain Reaction (qRT-PCR)

Twelve differentially expressed genes were selected for qRT-PCR analysis (Appendix A). The Oligo7 (version 7.56) software was used to design primers. PrimerScript RT reagent kit with a gDNA eraser (Perfect Real Time) (TaKaRa, RR047A) was used to synthesize cDNA according to the manufacturer’s protocol. Three technical replicates were performed for each sample. The final reaction volume of 10 µL contained 2 µL of cDNA, 8 µL of qPCR master mix (5 µL of TB Green Premix Ex Taq II system according to the manufacturer’s instructions (TaKaRa, RR820A), 0.25 µL of each primer, and 2.75 µL of RNA-free water). The samples were run on a CFX96 Touch Real-Time System (Bio-Rad, Hercules, CA, USA). The conditions were 94 °C for 30 s, followed by 39 cycles of 94 °C for 5 s and 65 °C for 34 s. The relative mRNA levels of the candidate key genes were calculated using the 2^–ΔΔCt^ method.

### 4.10. Pathway Enrichment Analysis

The differential expression genes (DEGs) result was selected based on an adjusted *p* value < 0.05. These genes were analyzed using the DAVID Bioinformatics Resource 6.8 (https://david.ncifcrf.gov/ (accessed on 20 May 2022)) and clusterprofiler 4.0. According to gene function and pathway annotation, KEGG and GO graphs were designed using GraphPad Prism 7 (version 7.0.1) software.

### 4.11. Cell Culture

Porcine kidney cells (PK15 cells) were preserved in the house. Porcine skeletal muscle satellite cells (PSC cells) were purchased from iCell Bioscience Inc. (Shanghai, China). PK15 cells were maintained in Dulbecco’s modified eagle media (DMEM) (Gibco, Thermo Fisher Scientific, Shanghai, China) supplied with 10% fetal bovine serum (FBS) (Gibco, Thermo Fisher Scientific, Shanghai, China) and 1% penicillin/streptomycin (Thermo Fisher Scientific, Shanghai, China). PSC cells were maintained in skeletal muscle cell complete medium (PriMed-iCell-018, iCell Bioscience Inc., Shanghai, China) supplied with 10% FBS (iCell Bioscience Inc., Shanghai, China), 1% penicillin/streptomycin (iCell Bioscience Inc., Shanghai, China), and 1% additive (iCell Bioscience Inc., Shanghai, China). The cells were cultured in a 37 °C incubator with 5% CO_2_.

### 4.12. Transwell Assay

Small interfering RNA (siRNA) for pig siACBD7 # 1 (GGAACTGAAAGAACTCTAT), siACBD7 # 2 (GCCATGAGTGCCTATATTT), and a control siRNA (1022076) were obtained from RIBOBIO (Guangzhou RiboBio Co., Ltd., Guangzhou, China). The siRNA was transfected using Lipofectamine™ RNAiMAX Transfection Reagent (Invitrogen, Waltham, MA, USA). Briefly, 9 µL of Lipofectamine RNAiMax were diluted with 150 µL OPTI-MEM, and 3 µL of 10 µmol/L siRNA were diluted with 150 µL OPTI-MEM. Diluted Lipofectamine RNAiMax and siRNA were then mixed 1:1 and incubated for 5 min at normal temperature. After incubation, the mixture was plated on a 3.5 cm dish with 2 mL OPTI-MEM. The cells were incubated for 1–3 days at 37 °C and then analyzed to identify transfected cells. PK15 cells were transfected for 24 h, 48 h, and 72 h, and the fluorescence expression was observed under a fluorescence microscope. The fluorescence intensity was analyzed and recorded using Image J software. To validate the function of *ACBD7* on cell proliferation, the *ACBD7* overexpression vector was transfected into the pig SkMCs cells. The cells were stained with 10 µg/mL DAPI for 10 min, and the proliferation cells in three random fields were counted under a fluorescence microscope (ZEISS, Jena, Germany).

### 4.13. Statistical Analysis

The results of qRT-PCR were statistically analyzed, and a line chart was drawn using the GraphPad Prism 7 software. The data of experiments were expressed as the mean ± standard deviation of three independent experiments, and comparisons between groups were first tested for normal distribution using the Shapiro–Wilk test, then parametric testing using the Brown–Forsythe test, and finally a one-way ANOVA with Tukey’s post hoc test. *p* < 0.05 was considered statistically significant. Analysis of the above data as well as PCA and heat map functions were all implemented using R 3.5.0 (http://www.r-project.org/(accessed on 24 February 2022)). A univariate analysis of variance (ANOVA) was used to determine the significance of differences in relative contents between different groups. SPSS19.0 software was used to analyze the fluorescence expression of the disturbed ACBD7 gene in PK15 cells.

## 5. Conclusions

In this study, we provide a new avenue for understanding the transcriptome and accessible chromatin dynamics during Landrace pigs skeletal muscle development after birth. The RNA-seq analysis identified 8554 effective differential genes, among which ACBD7 was identified as one of the key genes related to the development of porcine skeletal muscle. Some potential cis-regulatory elements identified by ATAC-seq analysis contain binding sites for many transcription factors, including SP1 and EGR1. In summary, comprehensive high-resolution gene expression maps were developed for the transcriptome and accessible chromatin during postnatal skeletal muscle development in pigs.

## Figures and Tables

**Figure 1 ijms-24-06413-f001:**
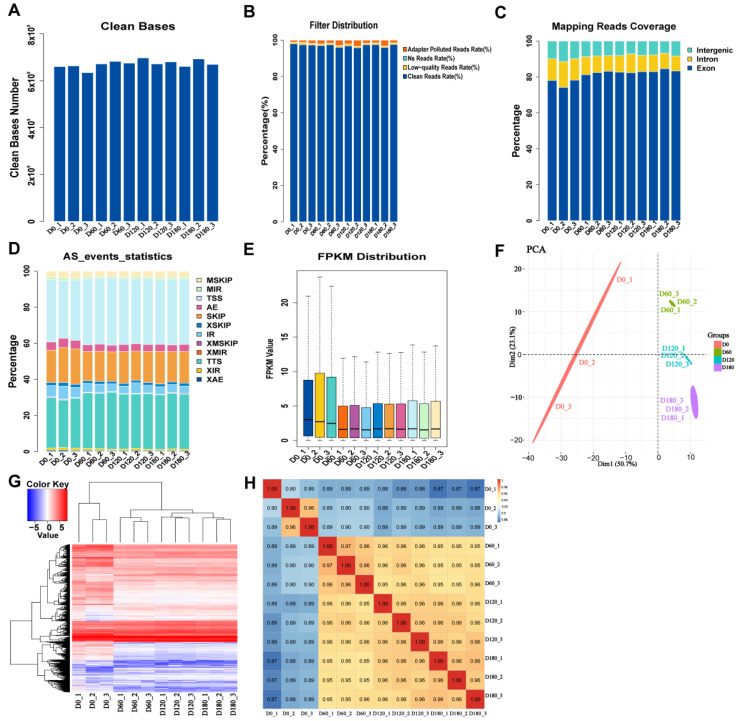
Data filtering, alignment analysis, and expression statistics for RNA-seq. (**A**) Distribution map of clean bases for all samples. (**B**) A bar plot of filter distribution for all samples. (**C**) The distribution of unique aligned sequences in each region of the reference genome. (**D**) A bar plot of alternative splicing results. (**E**) Expression box distribution. (Note: The abscissa is the sample name, and the ordinate is the logarithm of gene expression in each sample. From top to bottom are the maximum, upper quartile, median, lower quartile, and minimum). (**F**) Principal components analysis (PCA) plot. (**G**) Heatmap of the distribution of gene expression levels in each sample. (**H**) Heatmap showing the Pearson correlation results.

**Figure 2 ijms-24-06413-f002:**
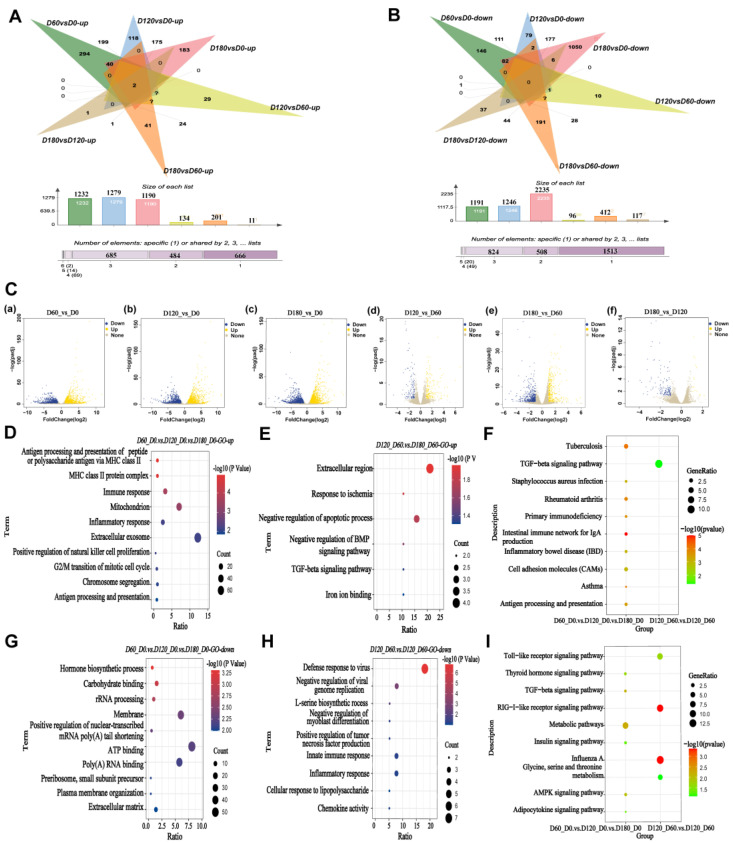
Enrichment analysis of differentially expressed genes during postnatal skeletal muscle growth. (**A**) Statistical graph of significantly different genes up-regulated in D60 vs. D0, D120 vs. D0, D180 vs. SD0, D120 vs. D60, D180 vs. D60, and D180 vs. D120 in four periods of Landrace pig skeletal muscle pairwise comparison |log2 (fold change)| ≥ 1 and FDR < 0.05). (**B**) Statistical graph of significantly different genes down-regulated in D60 vs. D0, D120 vs. D0, D180 vs. SD0, D120 vs. D60, D180 vs. D60, and D180 vs. D120 in four periods of Landrace pig skeletal muscle pairwise comparison (|log2 fold change ≥ 1| and FDR < 0.05). (**C**) (**a**) Volcano map for D60 vs. D0 group; (**b**) Volcano map for D120 vs. D0 group; (**c**) Volcano map for D180 vs. D0 group; (**d**) Volcano map for D120 vs. D60 group; (**e**) Volcano map for D180 vs. D60 group; (**f**) D180 vs. D120 group. (**D**) Top 10 up-regulated entries enriched using GO for differentially co-expressed genes in D60 vs. D0, D120 vs. D0, and D180 vs. D0 groups. (**E**) Top 10 up-regulated entries enriched using GO for differentially co-expressed genes in D120 vs. D60, D180 vs. D60 groups. (**F**) Top 10 up-regulated entries enriched using KEGG for differentially co-expressed genes in D60 vs. D0, D120 vs. D0, and D180 vs. D0 groups and D120 vs. D60, D180 vs. D60 groups. (**G**) Top 10 down-regulated entries enriched using GO for differentially co-expressed genes in D60 vs. D0, D120 vs. D0, and D180 vs. D0 groups. (**H**) Top 10 down-regulated entries enriched by GO for differentially co-expressed genes in D120 vs. D60, D180 vs. D60 groups. (**I**) Top 10 down-regulated entries enriched using KEGG for differentially co-expressed genes in D60 vs. D0, D120 vs. D0, and D180 vs. D0 groups and D120 vs. D60, D180 vs. D60 groups.

**Figure 3 ijms-24-06413-f003:**
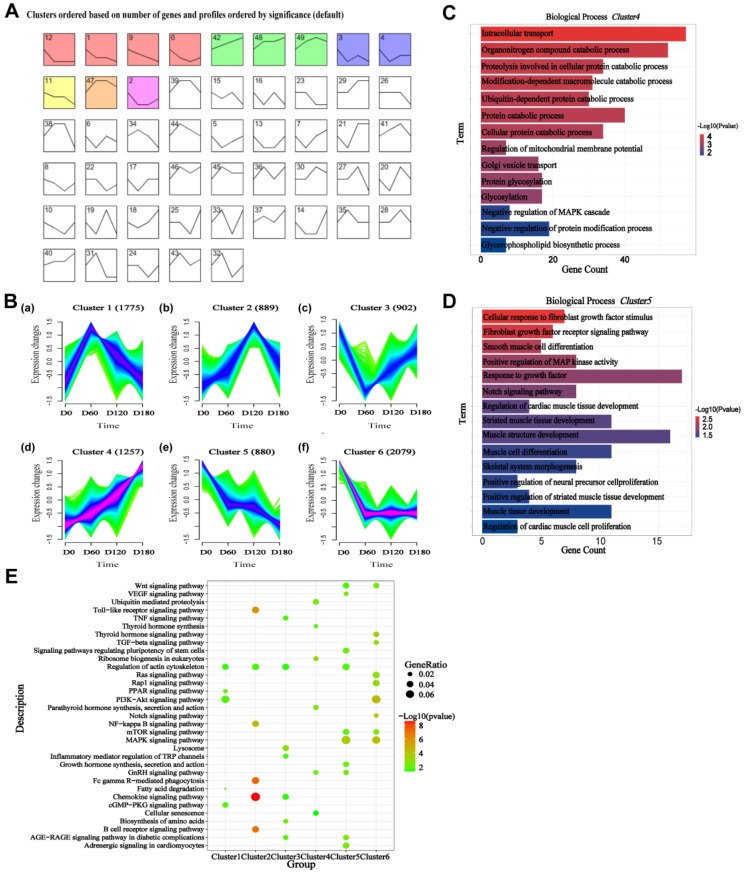
Trend analysis for all RNA−seq genes, cluster analysis, and gene ontology. (**A**) STEM clustering of skeletal muscle during skeletal muscle development in Landrace pigs. Colored profiles indicate that the genes in the module are significantly different (*p* < 0.05), and profiles without color indicate that the genes in the module are not significantly different. The same colors represent similar trends. (**B**) Clustering of the genes based on their expression patterns at D0, D60, D120, and D180. The numbers in parentheses indicate the number of genes in a cluster. The *x*-axis represents the sample, and the *y*-axis represents centralized and normalized expression values. The black lines are the mean expression trends of the genes within each cluster. (**C**) The biological process map of the gene set cluster4 that was up-regulated during the development of Landrace pig skeletal muscle through GO enrichment analysis. Significant categories are displayed. (**D**) The biological process map of the gene set cluster5 that was down-regulated during the development of Landrace pig skeletal muscle through GO enrichment analysis. Significant categories are displayed. (**E**) The six trend cluster genes obtained using the Mffuz package analysis were subjected to KEGG enrichment.

**Figure 4 ijms-24-06413-f004:**
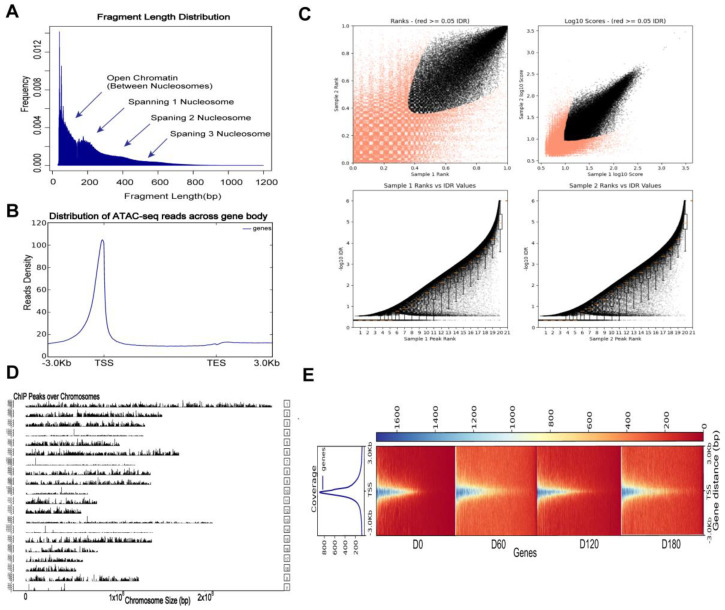
ATAC−seq read mapping, processing, and visualization. (**A**) The distribution of insert fragments in Landrace. (**B**) Mapped read distributions (from bigwig) across gene bodies and peaks. The *x*-axis represents the normalized gene or peak length, and the *y*-axis represents the read enrichment. The larger the value, the more enriched. TSS stands for the gene start site, and TES stands for the gene stop site. (**C**) Concordance plot of peaks between replicate samples evaluated using IDR. Top left: Rep1 peak ranks vs. Rep2 peak ranks, with peaks not passing a specific IDR threshold shown in red. Top right: Rep1 log10 peak scores vs. Rep2 log10 peak scores; peaks that do not pass a specific IDR threshold are shown in red. The following two Figures: Peak rank vs. IDR scores, the boxplot shows the distribution of IDR values, by default, the threshold of IDR values is −1 × 10^−6^). (**D**) The dynamic changes of global chromatin accessibility. (**E**) The heatmap of the peak signals across the gene body of library in Landrace. −3.0 represents 3 kb of TSS, the Deeptools tool plotHeatmap was used for this analysis.

**Figure 5 ijms-24-06413-f005:**
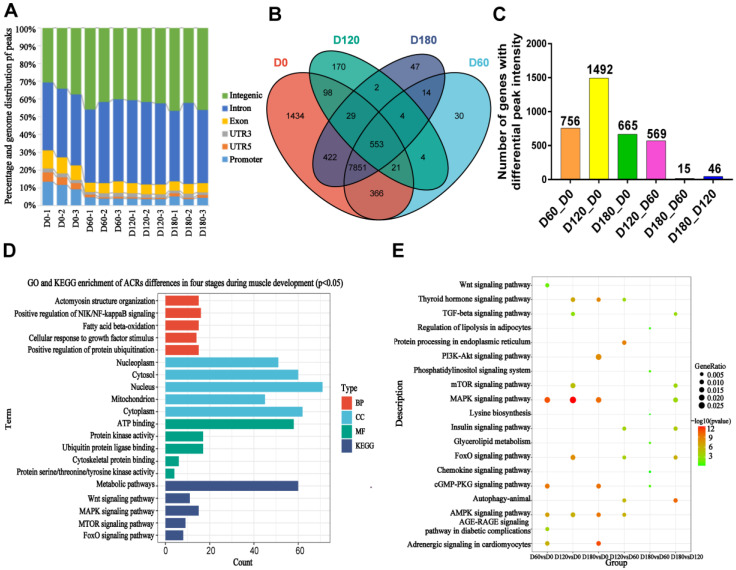
Identification and genomic distribution of muscle tissue accessible chromatin regions. (**A**) Genomic distribution of ATAC-seq peaks (i.e., mapped reads) in each of the muscle samples that were analyzed using ATAC-seq. Each genomic region is labeled and color-coded, as indicated above the stacked bar chart. The genome-wide distribution of the peaks. The genome-wide functional regions were divided into promoter, 3′UTR, 5′UTR, coding exon, intron, and distal intergenic regions. (**B**) Venn diagram of ACR common to the D0, D60, D120, and D180 stages. (**C**) Differential ACRs during muscle development were annotated to genes at D60 vs. D0, D120 vs. D0, D180 vs. D0, D120 vs. D60, D180 vs. D60, and D180 vs. D120 stages. (**D**) GO and KEGG enrichment of ACR differences in four stages during muscle development (*p* < 0.05). (**E**) KEGG-enriched entries for genes annotated through differential ACR in the D60 vs. D0, D120 vs. D0, D180 vs. D0, D120 vs. D60, D180 vs. D60, and D180 vs. D120 groups.

**Figure 6 ijms-24-06413-f006:**
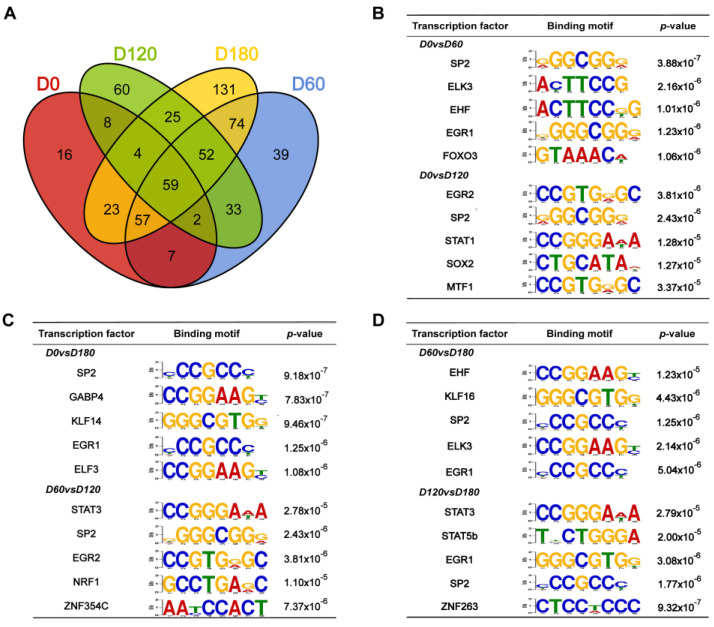
Motif analysis of differential peaks during postnatal porcine skeletal muscle development. (**A**) Venn diagram of motif differentials of D0, D60, D120, and D180 during postnatal skeletal muscle development in pigs. (**B**–**D**) Top five transcription factor binding motifs enriched in significantly peak regions according to the *p*-values between D0 vs. D60, D0 vs. D120, D0 v D120, D60 vs. D120, D60 vs. D180, and D120 vs. D180, respectively.

**Figure 7 ijms-24-06413-f007:**
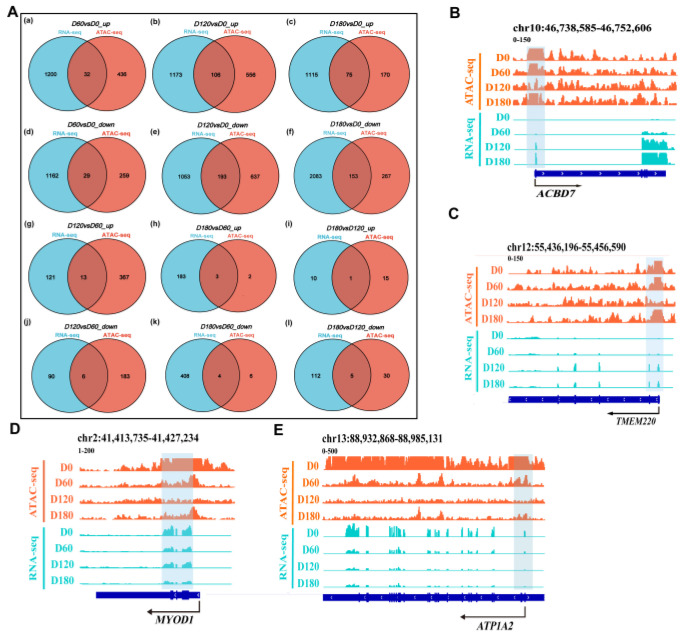
The results of ATAC-Seq and RNA-Seq were integrated. (**A**) Overlap of significantly expressed genes identified using RNA-seq with nearby genes of significantly increased ATAC-seq peaks between D60 vs. D0, D120 vs. D0, D180 vs. D0, D120 vs. D60, D180 vs. D60, and D180 vs. D120 groups, respectively. (**a**) Venn diagram of D60 vs. D0_up groups; (**b**) Venn diagram of D120 vs. D0_up groups; (**c**) Venn diagram of D180 vs. D0_up groups; (**d**) Venn diagram of D60 vs. D0_down groups; (**e**) Venn diagram of D120 vs. D0_down groups; (**f**) Venn diagram of D180 vs. D0_down groups; (**g**) Venn diagram of D120 vs. D60_up groups; (**h**) Venn diagram of D180 vs. D60_up groups; (**i**) Venn diagram of D180 vs. D120_up groups; (**j**) Venn diagram of vs. D120_D60 down groups; (**k**) Venn diagram of D180 vs. D60_down groups; (**l**) Venn diagram of D180 vs. D120_down groups. (**B**) IGV snapshot of Acyl-CoA Binding Domain Containing 7 (*ACBD7*). (**C**) IGV snapshot of Transmembrane Protein 220 (*TMEM220*). (**D**) IGV snapshot of Myogenic Differentiation 1 (*MYOD1*). (**E**) IGV snapshot of *ATP1A2*, the gene that encodes the alpha-2 subunit of the Na^+^/K^+^ ATPase.

**Figure 8 ijms-24-06413-f008:**
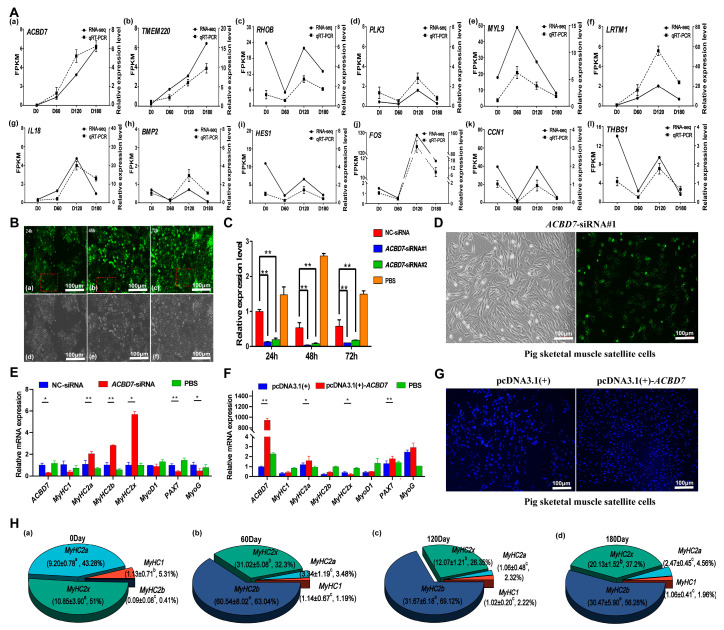
Functional validation of candidate DEGs. (**A**) The line chart of qRT-PCR and RNA-seq data for the following genes: (**a**) *ACBD7* gene; (**b**) *TMEM220* gene; (**c**) *RHOB* gene; (**d**) *PLK3* gene; (**e**) *MYL9* gene; (**f**) *LRTM1* gene; (**g**) *IL18* gene; (**h**) *BMP2* gene; (**i**) *HES1* gene; (**j**) *FOS* gene; (**k**) *CCN1* gene; (**l**) THBS1 gene. The qRT-PCR results are expressed as the mean ± SEM of three skeletal muscles per group and represented by the dashed line, while the RNA-seq data are represented by the solid line. (**B**) Fluorescence effect of 24, 48, and 72 h after transfection of siACBD7 in PK15 cells. (**a**) Green fluorescence image of PK15 cells transfected for 24 h. (**b**) Green fluorescence image of PK15 cells transfected for 48 h. (**c**) Green fluorescence image of PK15 cells transfected for 72 h. (**d**) White light images of PK15 cells transfected for 24 h. (**e**) White light images of PK15 cells transfected for 48 h. (**f**) White light images of PK15 cells transfected for 72 h. Note: The red square indicates representative specific fluorescence. (**C**) The relative expression levels after 24, 48, and 72h of transfection of siACBD7 in PK15. (**D**) The fluorescence effect of 48 h after transfection of *ACBD7*-siACBD7#1 in porcine skeletal muscle satellite cells. (**E**,**F**) It interferes with the relative expression levels of *ACBD7*, *MyHC1*, *MyHC2a*, *MyHC2b*, *MyHC2x*, *MyOD1*, *PAX7*, and *MyOG* genes in pig skeletal muscle satellite cells transfected with the *ACBD7* gene. Note: * means significant difference (*p* < 0.05); ** means very significant difference (*p* < 0.01) (**G**) Transwell assay showing the change in muscle proliferation after overexpressing *ACBD7* in porcine skeletal muscle satellite cells. (**H**) Dynamic changes of *MyHC1*, *MyHC2a*, *MyHC2x*, and *MyHC2b* in skeletal muscle of Landrace pigs at the 0, 60, 120, and 180 days. Note: a,b,c indicates data in the same row with completely different lowercase letters indicates significant difference (*p* < 0.05), and data with any same lowercase letters or no letters indicates no significant difference (*p* > 0.05).

## Data Availability

The ATAC-Seq and RNA-Seq data in this study have been deposited in the NCBI Gene Expression Omnibus (GEO) under the accession number PRJNA883228.

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
