# Peer review of "The Landscape of Accessible Chromatin and Developmental Transcriptome Maps Reveal a Genetic Mechanism of Skeletal Muscle Development in Pigs"

_ijms, 2023, doi:10.3390/ijms24076413_

Round 1
Reviewer 1 Report
1. The article introduced some studies about embryonic skeletal muscle at 33, 65, and 90 days of age. But it does not explain the reason why you select 0 d, 60 d, 120 d, and 180 d of porcine skeletal muscleas samples for your study.
2. How to explain the large difference in distance between D0 groups in the dendrogram results. (Supplemental Fig. S2.)
3. It is necessary to add different parts of skeletal muscle as controls when studying the regulatory effect of the ACBD7 gene on skeletal muscle satellite cells. (Line 300~304)
4. Fig.8B cannot reflect that the effect of 48-h transfection is best, and it is necessary to add the statistics and analysis of fluorescence intensity. (Line 305~308)
5. There is a citation error in the table. (Line 103)
Author Response
Dear Editors and Reviewers:
Thank you for your letter and for the reviewers’ comments concerning our manuscript entitled “The landscape of accessible chromatin and developmental transcriptome maps reveal a genetic machanism of skeletal muscle development in pigs.” (ID: 2277085). Those comments are all valuable and very helpful for revising and improving our paper, as well as the important guiding significance to our researches. We have studied comments carefully and have made correction which we hope meet with approval. Revised portion are marked in red in the paper. The main corrections in the paper and the responds to the reviewer’s comments are as flowing:
Responds to the reviewer’s comments:
Reviewer #1:
- Response to comment:The article introduced some studies about embryonic skeletal muscle at 33, 65, and 90 days of age. But it does not explain the reason why you select 0 d, 60 d, 120 d, and 180 d of porcine skeletal muscles samples for your study.
Response: Embryonic skeletal muscle in the early stage of development is most widely studied. However, there are relatively few studies on pig postnatal skeletal muscle, and even fewer studies on the landscape of accessible chromatin in pig postnatal skeletal muscle development. At present, one of the main lean commercial pigs on the market is Landrace. In order to explore its meat production traits and the mechanism of skeletal muscle development, from the perspective of production practice. So, we have chosen 0 d, 60 d, 120 d, and 180 d of porcine skeletal muscles samples for our study. Considering the Reviewer’s suggestion, we have made changes. Line 46-48,“The results revealed the differences between the breeds in terms of their molecular mechanisms for skeletal muscle growth and development.” was deleted. “However, there were relatively few studies on pig postnatal skeletal muscle. In addition,” was added.
- Response to comment: How to explain the large difference in distance between D0 groups in the dendrogram results. (Supplemental Fig. S2.)
Response: The early stage of animal growth is an important period of muscle fiber metabolism and contraction type transformation. This stage is an important turning point for pigs from embryo to birth, especially for 0-day-old pigs, and it may also be an active period of gene transcription or chromatin accessibility.
- Response to comment:It is necessary to add different parts of skeletal muscle as controls when studying the regulatory effect of the ACBD7 gene on skeletal muscle satellite cells. (Line 300~304)
Response: We have made correction according to the Reviewer’s comments. Total RNA was extracted from three Landrace pigs’ leg muscles at day 0 and 180 using the traditional Trizol method. The samples were run on CFX96 Touch Real-Time System ( Bio-Rad ). Finally, the software GraphPad Prism 7 was used to combine the results with the previous results. The results were shown in Supplemental Fig. S10.
- Response to comment: Fig.8B cannot reflect that the effect of 48-h transfection is best, and it is necessary to add the statistics and analysis of fluorescence intensity. (Line 305~308)
Response: Considering the Reviewer’s suggestion, we have made changes. PK15 cells were transfected for 24h, 48h and 72h, and the fluorescence expression was observed under fluorescence microscope. The fluorescence intensity was analyzed and recorded using Image J software. SPSS19.0 software was used to analyze the fluorescence expression of the disturbed ACBD7 gene in PK15 cells. The results showed that the average fluorescence values at 48 h and 72 h were larger than those at 24 h. The average fluorescence values of 48 h and 72 h were not significant (p > 0.05). Combined with the results of qRT-PCR, it can be inferred that the effect of 48 h transfection is best. The results were shown in Supplemental Table S4.
- Response to comment: There is a citation error in the table. (Line 103)
Response: It is really true as Reviewer suggested that there is a citation error in the table. We carefully checked the table and the original content, “22 million reads” corrected as “44 million reads”.
We tried our best to improve the manuscript and made some changes in the manuscript. These changes will not influence the content and framework of the paper.
We appreciate for Editors and Reviewers’ warm work earnestly, and hope that the correction will meet with approval.
Once again, thank you very much for your comments and suggestions.

Reviewer 2 Report
This manuscript presents investigations of pig skeletal muscle development via RNA sequencing and chromatin accessibility profiling. Although this text is rich in experimental data and covers important knowledge gaps, the presentation of the results needs to be improved and the conclusions need refinement.
1. The results should be described making sure that the reader grasps the illustrations, but it is useless to enumerate what is already present in the figures. For example, the text between lines 141-148 is redundant. On the other hand, certain data are too laconically presented; for example, Clusters 1, 2, 3, and 6 from Fig. 3 are not described, and panels D and E of Fig. 4 are not analyzed except for the methods used to create them.
2. Subsection 4.13 needs careful revision. Please state clearly which methods of statistical analysis were used and for what purpose. For instance, Pearson's correlation coefficient is not a measure of statistical significance, as stated on line 567.
3. The figures need to be included in the main text, not as Supplementary Material. Moreover, they should be properly sized to enable the reader to decipher all the annotations. For example, Fig. 2C, Fig. 4D, and Fig. 4E have been saved at relatively low resolution, so that they are hard to read even if they are zoomed in. Also, the MDPI format asks for a lowercase labeling of figure panels (a), (b), ... instead of A, B, ....
4. The Supplementary Materials should be described briefly (i.e. by the number and title of each item) right after the main text of the paper (line 571). Explanations are given in the journal's Instructions for Authors (https://www.mdpi.com/journal/ijms/instructions) and the Microsoft Word template file, "ijms-template.dot".
5. Although a Conclusions section is not compulsory, it is highly recommended for such a complex study. It is up to the authors to decide whether they wish to include a separate section for their concluding remarks, but even if they decide not to do so, the final paragraph of Discussion should present specific conclusions drawn from the reported results. As it stands, a brief summary is given on lines 399-401, and future studies are outlined in the next 3 lines.
6. Minor revisions (original text => proposed revision):
line 2: Please use the right style for the entire title (the last 10 words are written in a different type).
Caption of Fig. S4: Venn diagram of three groups => Venn diagram of two groups
line 102: Remove one of the two periods after Fig 1A).
line 107: The name "histogram" is not appropriate for panels B, C, and D of Fig. 1. I would rather call them bar plots. A histogram plots the frequency distribution of data points of a given variable, according to a set of range groups (bins).
line 103: Supplemental Table S1 => Supplementary Table S1
line 114: (Fig. 1E and Supplemental S1) => (Fig. 1E and Supplementary Fig. S1)
line 117: Here, and in the following occurrences, please delete "Supplemental". The reader already knows that S1, S2, ... refer to Supplementary Materials.
line 164: will continue => continue
line 282: we used IGV software was used => we used the IGV software
Caption of Fig. 7E:
of ATPase Na+/K+ transporting subunit alpha 2 (ATP1A2) => ATP1A2, the gene that encodes
the alpha-2 subunit of the Na+/K+ ATPase.
line 338: over four periods => at four time points
line 511: RPKM => Reads Per Kilobase Million (RPKM)
Author Response
Dear Editors and Reviewers:
Thank you for your letter and for the reviewers’ comments concerning our manuscript entitled “The landscape of accessible chromatin and developmental transcriptome maps reveal a genetic machanism of skeletal muscle development in pigs.” (ID: 2277085). Those comments are all valuable and very helpful for revising and improving our paper, as well as the important guiding significance to our researches. We have studied comments carefully and have made correction which we hope meet with approval. Revised portion are marked in red in the paper. Should you have any questions, please contact us without hesitate. Here below is our description on revision according to the reviewers’ comments.
Responds to the reviewer’s comments:
Reviewer #2:
- Response to comment:The results should be described making sure that the reader grasps the illustrations, but it is useless to enumerate what is already present in the figures. For example, the text between lines 141-148 is redundant. On the other hand, certain data are too laconically presented; for example, Clusters 1, 2, 3, and 6 from Fig. 3 are not described, and panels D and E of Fig. 4 are not analyzed except for the methods used to create them.
Response: We have re-written the results according to the Reviewer’s suggestion. The entire paragraph entitled “ 2.2. Enrichment analysis of differentially expressed genes during postnatal skeletal muscle growth” , “2.3. Trend analysis for all RNA-seq genes, cluster analysis, and Gene Ontology”, and “ATAC-seq read mapping, processing, and visualization” have been carefully and significantly revised. On the other hand, Clusters 1, 2, 3 and 6 in Fig. 3 were described, and supplementary figures S10, S11, S12 and S13 were drawn respectively. Next, we analyzed the panels D and E in Fig. 4 in addition to explaining the methods used to create them. See the red part of the original text.
- Response to comment: Subsection 4.13 needs careful revision. Please state clearly which methods of statistical analysis were used and for what purpose. For instance, Pearson's correlation coefficient is not a measure of statistical significance, as stated on line 567.
Response: We have made the following revisions according to the opinions of the reviewers. The results of qRT-PCR were statistically analyzed and a line chart was drawn using the GraphPad Prism 7 software. The data of experiments were expressed as the mean ± standard deviation of three independent experiments, and comparisons between groups were first tested for normal distribution using the Shapiro–Wilk test, then parametric testing using the Brown–Forsythe test, and finally, a one-way ANOVA with Tukey’s post hoc test. p < 0.05 was considered statistically significant. Analysis of the above data as well as PCA and heat map functions were all implemented using R 3.5.0 (http://www.r-project.org/). A univariate analysis of variance (ANOVA) was used to determine the significance of differences in relative contents between different groups. SPSS19.0 software was used to analyze the fluorescence expression of the disturbed ACBD7 gene in PK15 cells.
- Response to comment:The figures need to be included in the main text, not as Supplementary Material. Moreover, they should be properly sized to enable the reader to decipher all the annotations. For example, Fig. 2C, Fig. 4D, and Fig. 4E have been saved at relatively low resolution, so that they are hard to read even if they are zoomed in. Also, the MDPI format asks for a lowercase labeling of figure panels (a), (b), ... instead of A, B, ....
Response: Considering the problem of with the typography of the articl and the main purpose of the article, all the figures are not included in the text, but some figures are used as supplementary materials. The size of the figures was adjusted appropriately, and we have used AI software to process the resolution of the image again. We have uploaded it in the revised article. In addition, we carefully read again the MDPI Style guide (https://www.mdpi.com/authors/layout#_bookmark83) and found that there were sentences that were described as follows: “For figures with more than one part, the panels should be labeled a, b, c, d, etc., and each part can be separately cited in the main text. Each part must be individually described in the caption. It is recommended to use label “A, B...” or “a, b...” instead of “left, right, top, bottom” for the subfigures.” Therefore, we carefully revised each graph. We have also uploaded it in the revised article.
- Response to comment: The Supplementary Materials should be described briefly (i.e. by the number and title of each item) right after the main text of the paper (line 571). Explanations are given in the journal's Instructions for Authors (https://www.mdpi.com/journal/ijms/instructions) and the Microsoft Word template file, "ijms-template.dot".
Response: Considering the Reviewer’s suggestion, we have briefly explained the number and title of each item of the supplementary material. As follows:
Fig.S1. Density distribution map of gene expression in each sample.
Fig.S2. Clustered dendrogram of 12 samples of skeletal muscle from Landrace pigs. Fig.S3 Venn diagram of three groups D60vsD0, D120vsD0, and D180vsS0 of landrace pig skeletal muscle.
Fig.S4. Venn diagram of two groups D180vsD60, D120vsD60, of landrace pig skeletal muscle.
Fig.S5. Venn diagram of three groups D60vsD0, D120vsD60, and D180vsD120 of landrace pig skeletal muscle.
Fig.S6. The biological process map of GO enrichment analysis after all genes in Profiles 12, 1, 9, and 0 was obtained after trend analysis during skeletal muscle development in Landrace pigs.
Fig.S7. The biological process map of GO enrichment analysis after all genes in Profiles 42, 48, 49 and 0 was obtained after trend analysis during skeletal muscle development in Landrace pigs.
Fig.S8. The biological process map of GO enrichment analysis after all genes in Profiles 3 and 4 was obtained after trend analysis during skeletal muscle development in Landrace pigs.
Fig.S9. The biological process map of GO enrichment analysis after all genes in Profiles 47 was obtained after trend analysis during skeletal muscle development in Landrace pigs.
Fig.S10. The biological process map of GO enrichment analysis after all genes in Cluster 1 was obtained after trend analysis during skeletal muscle development in Landrace pigs.
Fig.S11. The biological process map of GO enrichment analysis after all genes in Cluster 2 was obtained after trend analysis during skeletal muscle development in Landrace pigs.
Fig.S12. The biological process map of GO enrichment analysis after all genes in Cluster 3 was obtained after trend analysis during skeletal muscle development in Landrace pigs.
Fig.S13. The biological process map of GO enrichment analysis after all genes in Cluster 6 was obtained after trend analysis during skeletal muscle development in Landrace pigs.
Fig. S14. Expression of ACBD7 Gene in Different Tissues of Landrace pig.
Table S1. The summary of data generated by RNA sequencing.
Table S2. ATAC-seq data statistics.
Table S3. Intergrative RNA-seq and ATAC-seq analysis of GO and KEGG enrichment results for overlapping genes in D60vsD0, D120vsD0, D180vsDD0, D120vsD60, D180vsD60 and D180vsD120 groups.
Table S4. The fluorescence intensity of PK15 cells transfected for 24 h, 48 h, and 72 h was analyzed using Image J software.
Table S5. The primer sequences for RT-PCR.
- Response to comment: Although a Conclusions section is not compulsory, it is highly recommended for such a complex study. It is up to the authors to decide whether they wish to include a separate section for their concluding remarks, but even if they decide not to do so, the final paragraph of Discussion should present specific conclusions drawn from the reported results. As it stands, a brief summary is given on lines 399-401, and future studies are outlined in the next 3 lines.
Response: It is really true as Reviewer suggested that we need a Conclusions section. We have included a separate section in the concluding comments. As follows:
In this study, our study provides a new avenue for understanding the transcriptome and accessible chromatin dynamics during Landrace pigs skeletal muscle development after birth. The RNA-seq analysis identified 8554 effective differential genes, among which ACBD7 was identified as one of the key genes related to the development of porcine skeletal muscle. Some potential cis-regulatory elements identified by ATAC-seq analysis contain binding sites for many transcription factors, including SP1 and EGR1. In summary, comprehensive high-resolution gene expression maps were developed for the transcriptome and accessible chromatin during postnatal skeletal muscle development in pigs.
- Response to comment: Minor revisions (original text => proposed revision):
line 2: Please use the right style for the entire title (the last 10 words are written in a different type).
Response: Good advice, we have revised it.
Caption of Fig. S4: Venn diagram of three groups => Venn diagram of two groups
Response: We have revised it.
line 102: Remove one of the two periods after Fig 1A).
Response: We have revised it.
line 107: The name "histogram" is not appropriate for panels B, C, and D of Fig. 1. I would rather call them bar plots. A histogram plots the frequency distribution of data points of a given variable, according to a set of range groups (bins).
Response: Thank you for your suggestion, we have revised it.
line 114: (Fig. 1E and Supplemental S1) => (Fig. 1E and Supplementary Fig. S1)
Response: We have revised it.
line 117: Here, and in the following occurrences, please delete "Supplemental". The reader already knows that S1, S2, ... refer to Supplementary Materials.
Response: We have revised it.
line 164: will continue => continue
Response: Since the content of the original sentence is a bit wordy, we have deleted it here.
line 282: we used IGV software was used => we used the IGV software
Response: We have revised it.
Caption of Fig. 7E: of ATPase Na+/K+ transporting subunit alpha 2 (ATP1A2) => ATP1A2, the gene that encodes the alpha-2 subunit of the Na+/K+ ATPase.
Response: Good advice, we have revised it.
line 338: over four periods => at four time points
Response: We have revised it.
line 511: RPKM => Reads Per Kilobase Million (RPKM)
Response: We have revised it.
We tried our best to improve the manuscript and made some changes in the manuscript. These changes will not influence the content and framework of the paper.
We appreciate for Editors and Reviewers’ warm work earnestly, and hope that the correction will meet with approval.
Once again, thank you very much for your comments and suggestions.

Round 2
Reviewer 2 Report
The authors managed to properly take into account all of my comments regarding the previous version of the manuscript.
I fully agree that not all figures would fit in the main paper. The authors have chosen the most relevant ones and listed the titles of those left in the Supplementary Material.